# Predicting water status, growth and yield of tomato under different irrigation regimes using the RGB image indices and artificial neural network model

Mohamed S. Abd El-baki[1], Mohamed Maher Ibrahim[1], Salah Elsayed[2]*, Ahmed Elbeltagi[1]*, Ali Salem [3,4]*, Nadia G. Abd El-Fattah[1]

**1** Agricultural Engineering Department, Faculty of Agriculture, Mansoura University, Mansoura, Egypt, **2** Agricultural Engineering, Evaluation of Natural Resources Department, Environmental Studies and Research Institute, University of Sadat City, Minufiya, Egypt, **3** Civil Engineering Department, Faculty of Engineering, Minia University, Minia, Egypt, **4** Structural Diagnostics and Analysis Research Group, Faculty of Engineering and Information Technology, University of Pecs, Pecs, Hungary

* salem.ali@mik.pte.hu (AS); salah.emam@esri.usc.edu.eg (SE); ahmedelbeltagy81@mans.edu.eg (AE)

## Abstract

Water stress is a global challenge that severely impacts crop production by hindering essential physiological processes. To address this issue, proximal sensing has emerged as a promising technique for the early identification of stress in vegetables, enabling timely management interventions and optimizing yield. This study aimed to use RGB image indices and an artificial neural network (ANN) model to quantify the responses of various plant traits, such as fresh biomass (FB) weight, dry biomass (DB) weight, canopy water content (CWC), relative chlorophyll content (SPAD), soil moisture content (SMC), and tomato yield across different irrigation levels. Field experiments were conducted during the 2022 and 2023 growing seasons, capturing digital RGB images and measuring plant traits at the flowering and fruit-ripening stages. The results revealed that a reduced irrigation level led to a decrease in various plant traits. The study also revealed significant differences in RGB image indices between different irrigation levels, with strong positive relationships identified for the majority of RGB image indices incorporating green components (G) and $R^2$ reaching 0.99 for various plant traits. However, the red-blue simple ratio (RB) index, which does not consider the G, did not significantly correlate with any of the plant traits. The ANN models achieved high prediction accuracy, with high $R^2$ values reaching 0.99 for various plant traits and yields. These findings underscore the practicality and reliability of employing RGB imaging indices in conjunction with ANN models for effectively managing tomato crop growth and production, particularly under limited water conditions.

**Editor:** Ömer Faruk Coşkun, Mustafa Kemal University: Hatay Mustafa Kemal Universitesi, TÜRKIYE

**Data availability statement:** All data are presented within the article.

**Funding:** The author(s) received no specific funding for this work.

**Competing interests:** The authors have declared that no competing interests exist.

## 1. Introduction

The scarcity of water presents a formidable challenge for achieving sustainable agriculture in arid and semi-arid countries, especially considering the increasing unpredictability of climate patterns that greatly impact the agricultural sector [1]. Future forecasts indicate that water scarcity is likely to increase globally by up to 20% due to climate change. Arid and semi-arid areas are mainly dependent on irrigation by farmers. Irrigated land plays an important role in food security, accounting for 40% of the world's total food [2]. It is estimated that a large proportion of the available water, up to 25–40%, will be redistributed between different sectors, with the agricultural sector likely to bear the brunt as a result of consuming a large amount of water. However, the limited supply of irrigation water poses a serious threat to food security in the future. Therefore, it is necessary to develop strategies for water conservation and ensure the effective use of this resource by improving crop productivity per unit of water rather than per unit of land area [3].

In 2022, the local tomato harvest areas in Egypt covered approximately 143,618 hectares, yielding around 6.28 million tons of tomatoes [4]. Effective management of irrigation processes has an important role in the growth of tomato crops [5,6]. Numerous studies have confirmed that minor errors in the management of irrigation processes cause deterioration in crop growth, especially during sensitive growth stages. This deterioration can lead to obvious reductions in the productivity and quality of the crop, as well as the net profit from the production system [7]. Therefore, it is necessary to adopt appropriate irrigation techniques to improve the efficiency of water use under conditions of limited irrigation water while maintaining proper production of tomato crop [8].

For the effective management of deficit irrigation methods, various plant attributes such as fresh biomass (FB), dry biomass (DB), canopy water content (CWC), relative chlorophyll content (SPAD), crop yield can be used. These attributes are commonly used to assess the condition of crops under the stress of water shortage. However, the essence of the benefit learned from observing these features lies in the immediate and accurate assessment of the response of these attributes to water shortage [9]. Traditional techniques usually take a long time, material cost and considerable efforts to assess and monitor these plants attributes if used to collect field data on a large scale. Proximal remote sensing tools come as a solution to these problems to be exploited in proper management un-der deficit irrigation, as these tools provide accurate, fast and inexpensive assessment of the attributes of various plants.

Digital cameras can be used as an effective tool for proximal remote sensing in assessing and observing various features of a plant under water shortage [10]. These cameras capture an image of the plant canopy within the visible light spectrum, with a special focus on wavelengths ranging from 400 to 499 nm for blue, 500–549 nm for green, and 550–750 nm for red. By analyzing the data obtained from these three channels, various indices can be calculated. These indices provide sufficient information to estimate the various features of plants exposed to various stresses, including biotic and abiotic factors [11]. Notably, different plant traits such as SPAD, FB, DB, and CWC, which are difficult to assess visually, can be easily identified by changes in

the red, green, and blue channels captured using color digital cameras [12]. Although the cost of taking pictures of plants is cheap and fast to perform, the adoption of this technique to estimate the different features of plants on a large scale has been limited. Elsayed et al. [13] previously confirmed the possibility of using RGB images of a plant canopy to estimate the biomass and vegetative water content of a potato plant under water stress. However, the application of RGB image analysis techniques to assess the impact of different irrigation levels on various traits and yields in tomato plants, especially in arid conditions, remains limited.

When dealing with data gathered from proximate remote sensing instruments, it is crucial to handle it carefully because the data is often broad and lacks specificity, leading to significant overlap. In recent years, several artificial intelligence (AI) techniques have emerged as reliable and accurate solutions to address this issue. Artificial Neural Net-works (ANNs) are nonlinear mapping structures inspired by the human brain's functioning [14]. They possess the ability to learn and are designed to create mathematical models that mimic the computational power of the human brain. ANNs have proven their effectiveness in various practical applications [15]. In the agricultural domain, ANNs have been widely employed. They facilitate the development of models based on the inherent relationships between variables, eliminating the need for prior knowledge of their functional connections [16]. ANN models have been extensively utilized to predict diverse crop indicators, including growth, yield, and other biophysical processes for different vegetables such as tomato [17–19], lettuce [20,21], pepper [22,23], soybean [24], green peas [25], cabbage [26], onion [27], and potato [28].

Previous studies in the field have primarily concentrated on the utilization of RGB image indices derived from unsegmented color images, coupled with artificial neural networks or original color images employing conventional neural networks, to classify the water stress status of tomato plants [29,30]. Thus, to the best of our knowledge, no previous research has used artificial neural networks with the RGB image indices calculated from segmented color images to develop novel and robust models for the accurate estimation and monitoring of tomatoes' growth, chlorophyll content, water content, and yield at varying irrigation levels. Therefore, this study had three purposes: (i) to investigate the response of growth, chlorophyll content, water content, soil moisture content, and yield of tomatoes under different irrigation levels, (ii) to analyze the ability of RGB image indices to predict the traits of tomatoes indirectly across all the irrigation levels, growth stages, and growing seasons, and (iii) to determine the performance of ANN models based on RGB image indices.

## 2. Materials and methods

### 2.1. Experimental site, conditions, and design

Open-field experiments were carried out for two consecutive spring growing seasons during 2022 and 2023 on a private farm at Talkha, Dakahlia, Egypt (31.09° N, 31.38° E, and 17 meters elevation). During both growing seasons, the mean daily maximum temperature values were 46.1 and 42.3 °C, while the mean daily minimum temperature values were 7.2 and 9.3 °C, respectively. The highest mean relative humidity values were 79.3 and 68.5% in both seasons, while the lowest values were 35 and 25.9%, respectively. Regarding rainfall amounts, the second growing season recorded a higher value (24.20 mm) compared to the first growing season (20.0 mm). The soil texture was sandy clay (50.46% sand, 36.88% clay, and 12.66% silt). Its field capacity was 29.93%, and its wilting point was 14.6%. Soil and irrigation water had respective electrical conductivities of 0.74 and 0.83 dS/m, respectively.

The experimental design was a randomized, complete block with four replicates during both growing seasons. Irrigation regimes, namely, 50% (T50), 75% (T75), and 100% (T100) of full irrigation requirements, were applied by a drip irrigation. The experimental area covered 130 m². Each replicate consisted of one polyethylene (P.L.) lateral line with a 16-mm diameter. Each lateral line length was 9.0 m, and the emitter spacing was 0.40 m. There were 1.2 meters separating each lateral line, and it was connected to the sub-mainline (63 mm diameter, P.L.) via a 16 mm T-shaped plastic valve. This plastic valve controlled the irrigation depth at the desired level for each treatment separately. The emitters were built-in with an average discharge rate of 6 liter/hour at a 1 bar operating system. In the first season, tomato seeds 'Gs12 F1' hybrid was transplanted on February 23rd after a 35-day initial growth stage, and the harvest took place on June 17th. In the second

season, transplanting occurred on March 3rd, and harvesting took place on June 25th. Fertilizer requirements for the tomato crop were added according to the recommendations of the Egyptian Ministry of Agriculture. All treatments received 357 kg/ha of N (urea – 46.5% N), 60 kg/ha of P (phosphoric acid – 85% $P_2O_5$), and 238 kg/ha of K (potassium sulphate – 50% $K_2O$) through the drip irrigation system using a venturi-type injector in both seasons.

## 2.2. Irrigation water requirements

The CROPWAT model was suggested by Halimi et al. [31] and Gabr [32] to calculate the crop water requirement and decide when to irrigate. The FAO Penman-Monteith was used by this model as reported by Allen et al. [33]. The required meteorological data for experiment area were obtained from the NASA POWER | Data Access Viewer website: https://power.larc.nasa.gov/data-access-viewer/. to calculate reference evapotranspiration ($ET_o$). NASA POWER utilizes satellite observations, which can provide a broad view of global climate patterns. These observations contribute to the accuracy of the data, especially for regions with limited ground-based monitoring stations. Reanalysis datasets used in NASA POWER combine observations with numerical models to generate consistent long-term climate records. This approach enhances the accuracy and completeness of the data, as documented by Power [34].

For each growth stage, the crop coefficient ($K_c$) is multiplied by $ET_o$ to estimate crop evapotranspiration ($ET_c$). The total growing season for tomato plants was 150 days, divided into four stages: initial (35 days), development (39 days), middle (46 days), and late (30 days). The $K_c$ values were 0.38, 1.10, 1.10, and 0.65 during the initial, developmental, middle, and late stages, respectively, as reported by Noreldin et al. [35]. During the initial 15 days after transplanting (DAT), regular irrigation was employed for all treatments to make sure that optimal germination of seedlings occurred. Subsequently, the irrigation water applied adhered to the recommended quantity that was established for each irrigation treatment. Subsequently, these quantities were reduced by 25% and 50% for the T75 and T50, respectively.

## 2.3. Digital RGB imaging

### 2.3.1. Image acquisition and processing protocol using a digital camera.
The canopy of tomato plants in each treatment was captured using a 14-megapixel digital camera (Kodak D5100 reflex; Tokyo, Japan) between 11:00 and 13:00. The camera was fitted with a Sony 16–50 mm lens set at the minimum focal length, fixed aperture of F3.5, and shutter speed of 1/250. Because the photos were captured under clear-sky conditions, an umbrella was held above the camera to provide shade conditions. This was done to ensure homogeneous lighting and minimize the impact of direct sunlight and leaf reflections on image quality and pixel intensity in stereoscopic images. The camera had a resolution of 2454 × 2056 pixels, and it captured 16-bit RGB images (0.4-0.7 μm). The camera was manually held and positioned vertically overhead, maintaining a fixed distance of 0.8 meters from each tomato canopy plant. To ensure consistency, the flash was consistently turned off during the image capture process. To minimize angular distortion, the focal length was adjusted to 18 mm, providing a 76° image angle. After the digital photos were taken, it was saved in TIFF format and subjected to analysis with the OpenCV library and Python 3.11 software. There are occasionally non-canopy features in the pictures, such straw, weeds, and soil. As shown in Fig 1, image segmentation and extraction techniques were used to exclude any interference from non-canopy information throughout the feature extraction procedure.

### 2.3.2. Stages of image processing pipeline for vegetation extraction.
The pipeline involves several stages, including RGB channel separation, Color Index of Vegetation Extraction (CIVE), Otsu thresholding, masking, segmentation, and RGB color index calculation. In the first stage, the original image was split into its red, green, and blue channels. The second stage involved the calculation of the CIVE, a vegetation index that measures the difference between the green and other channels. This index can be used to identify areas of vegetation in the image [36]. Next, an Otsu threshold was applied to the CIVE image to segment it into foreground and background regions based on the distribution of pixel intensities. This step helped to further isolate the areas of the image that contain vegetation [37]. The generated binary picture was used as a mask for a masking operation that was applied to the original image after the

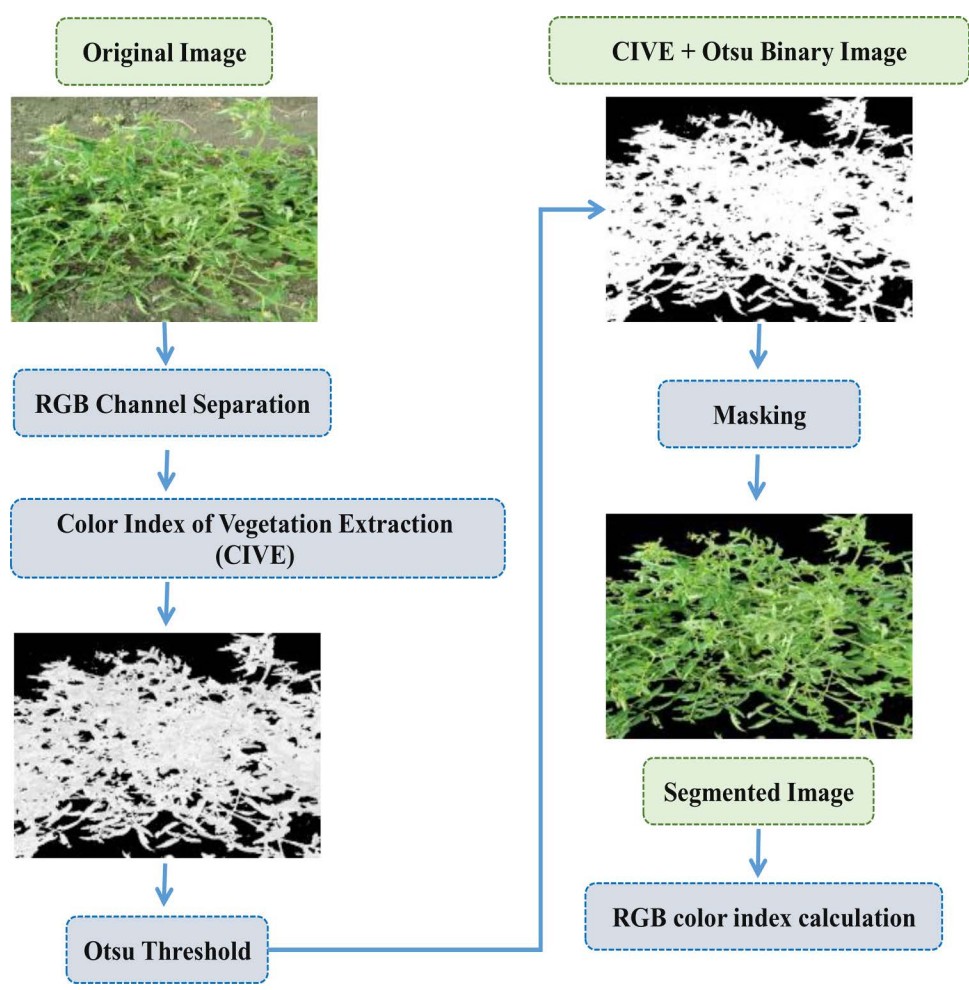

**Fig 1. Image processing pipeline for vegetation extraction.**

Otsu threshold has been applied. This operation essentially removes all non-vegetation areas from the original image, leaving only the parts that correspond to vegetation. The result of the masking operation was a segmented image, which showed only the areas of the original image that contain vegetation. This image can be useful for analyzing the distribution and characteristics of vegetation in the scene. Finally, several RGB color indices were calculated for the segmented image. RGB image was composed of three channels, which means that the color of each pixel can be represented by three values [38]. Numerous plant canopy-related parameters, including the amount of chlorophyll and the quantity of the green biomass, affect these values in the RGB channels. In order to extract sample features, the mean values of the RGB channels were obtained using the following equations:

$$R = \frac{1}{S_{num}} \sum_{i=1}^{S_{num}} R_i \qquad (1)$$

$$G = \frac{1}{S_{num}} \sum_{i=1}^{S_{num}} G_i \qquad (2)$$

$$B = \frac{1}{S_{num}} \sum_{i=1}^{S_{num}} B_i$$

(3)

where $R_i$, $G_i$, and $B_i$ stand for the digital image's red, green, and blue channels' respective pixel values. The initial pixel and the maximum number of pixels are denoted by the variables i and $S_{num}$, respectively. The average values of the red, green, and blue channels are denoted as R, G, and B, correspondingly. Table 1 contains the formulas and references for the twenty-one RGB image indices that were examined in the study.

## 2.4. Measurement of plant traits

The estimation of fresh biomass (FB), dry biomass (DB), and canopy water content (CWC) was meticulously assessed using the well-established procedures outlined by Semananda et al. [51]. After capturing RGB images of the canopies of tomato plants, biomass sampling was conducted on two times: in 2022, 67 and 93 days after transplanting (DAT), as well as twice in 2023 on 68 and 93 DAT. From each treatment, four tomato plants were randomly selected and cut at ground level from the imaged area. To find the FB weight, the recently cut plants were weighed right away. Following that, the plants were dried for

**Table 1. Various RGB image indices that were tested for this study.**

| RGB Image Indices | Formula | References |
|---|---|---|
| Red pixel percentage (R %) | $\frac{R}{R+G+B}$ | [38] |
| Green pixel percentage (G %) | $\frac{G}{R+G+B}$ | [38] |
| Blue pixel percentage (B %) | $\frac{B}{R+G+B}$ | [38] |
| Green red simple ratio (GR) | $\frac{G}{R}$ | [39] |
| Red blue simple ratio (RB) | $\frac{R}{B}$ | [39] |
| Green blue simple ratio (GB) | $\frac{G}{B}$ | [39] |
| Modified Green Red Vegetation Index (MGVRI) | $\frac{(G)^2-(R)^2}{(G)^2+(R)^2}$ | [39] |
| Red Green Blue Vegetation Index (RGVBI) | $\frac{G-(B \times R)}{(G)^2+(B \times R)}$ | [39] |
| Excess red vegetation index (ExR) | $(1.4 \times R) - G$ | [40] |
| Excess green vegetation index (ExG) | $(2 \times G) - R - B$ | [40] |
| Excess green minus Excess red index (ExGR) | $ExG - ExR$ | [40] |
| Vegetative index (VEG) | $\frac{G}{R^a \times B^{(1-a)}}$ , a = 0.667 | [41] |
| Color Index of Vegetation Extraction (CIVE) | $(0.441 \times R) - (0.811 \times G) + (0.385 \times B) + 18.78745$ | [42] |
| Combination (COM) | $(0.25 \times ExG) + (0.3 \times ExGR) + (0.33 \times CIVE) + (0.12 \times VEG)$ | [43] |
| Visible atmospherically resistant index (VARI) | $\frac{(G-R)}{(G+R+B)}$ | [44] |
| Green-red vegetation index (GRVI) | $\frac{(G-R)}{(G+R)}$ | [45] |
| Normalized difference index (NDI) | $128 \times [(\frac{G-R}{G+R}) + 1]$ | [46] |
| Triangular Greenness Index (TGI) | $G - (0.39 \times R) - (0.61 \times B)$ | [47] |
| Principal component analysis index (IPCA) | $0.994 \times (R - G) + 0.961 \times (G - B) + 0.914 \times (G - R)$ | [48] |
| Green Leaf Index (GLI) | $\frac{(2 \times G)-R-B}{(2 \times G)+R+B}$ | [49] |
| Modified Excess Green Index (MExG) | $(1.262 \times G) - (0.884 \times R) - (0.311 \times B)$ | [50] |

around 24 hours at 105°C in a forced-air oven. The DB weight was then estimated by weighing the dried plants. The following formula was used to determine the percentage of canopy water content (CWC) using the FB and DB weights:

$$CWC = \frac{FB - DB}{FB} * 100\%$$

(4)

For tomato fruit yield calculation, eight plants per treatment were randomly selected and labeled. These same plants were used to measure the yield during each pick up. The total tomato fruit yield for each treatment was calculated by averaging the weights of the eight selected plants, with two plants per replicate. The data were presented in tons per hectare.

The SPAD-502 chlorophyll meter was also used in this investigation to measure the relative chlorophyll content. Each plant was given four leaves, and the average reading was noted as the plant's representation. The SPAD readings were taken on the newest fully expanded leaf of all plants, following a specific location described by Bai and Purcell [52]. While, the soil samples were collected at 15 cm depths. SMC was estimated using the following equation according to [8,53].

$$SMC\ (\%) = \left( \frac{M_{wet} - M_{dry}}{M_{wet}} \right) * 100\%$$

(5)

Where:

SMC: Soil moisture content, %; $M_{wet}$: Weight of container with moist soil, g; $M_{dry}$: Weight of container with oven dry soil, g.

## 2.5. Artificial neural networks modeling

The artificial neural network (ANN) model was developed using the MLPRegressor class from the scikit-learn library (version 1.8.0) in Python to predict the following target variables: FB weight, DB weight, CWC, SMC, SPAD value, and tomato yield.

**2.5.1. Feature selection.** Before model development, input features (RGB image indices) were selected based on their correlation with the target variable, as illustrated in Fig 2. Only features with a correlation coefficient exceeding a predetermined threshold were retained. The optimal threshold was identified during the training phase by evaluating model performance and predictive efficiency across various threshold values. The final model was then validated using a separate test dataset.

The optimal R² threshold for feature selection was determined separately for each growth stage and target variable. During the flowering stage, features were selected using R² thresholds of 0.75, 0.75, 0.65, 0.65, 0.5, and 0.7 for FB, DB, CWC, SPAD, SMC, and Yield, respectively. During the fruit ripening stage, the corresponding selection thresholds were 0.9, 0.75, 0.85, 0.75, 0.7, and 0.7. For models trained on data from both stages combined, the thresholds used were 0.8, 0.75, 0.8, 0.1, 0.65, and 0.65 for the same variables.

**2.5.2. Data preprocessing and splitting.** The dataset was randomly partitioned into two independent subsets: a training set (70%) for model development and a test set (30%) reserved exclusively for evaluating the model's generalization performance, following established practices [54,55]. All input features were subsequently normalized using the StandardScaler method, which standardizes features by removing the mean and scaling to unit variance [56].

**2.5.3. Model architecture and training.** Hyper-parameters were predetermined before the training process of the model rather than being learned from the data. It was essential in determining the model's performance [57]. To achieve the best performance and generalization capability of an ANN model, grid-search hyper-parameter tuning was employed. Different combinations of hyper-parameters were systematically explored, including the hidden layer numbers (ranging from 1 to 5), the neuron numbers per hidden layer (ranging from 2 to 10), learning rate (fixed at 0.001), maximum iteration (600, 700, 800, 900, 1000), and activation functions (refer to Table 2 for specific functions used, as described by Sharma et al. [58]. Several solver algorithms were also evaluated for minimizing the loss function, namely stochastic gradient descent (SGD), the Quasi-Newton method (QN), and adaptive moment estimation (ADAM) [59–61].

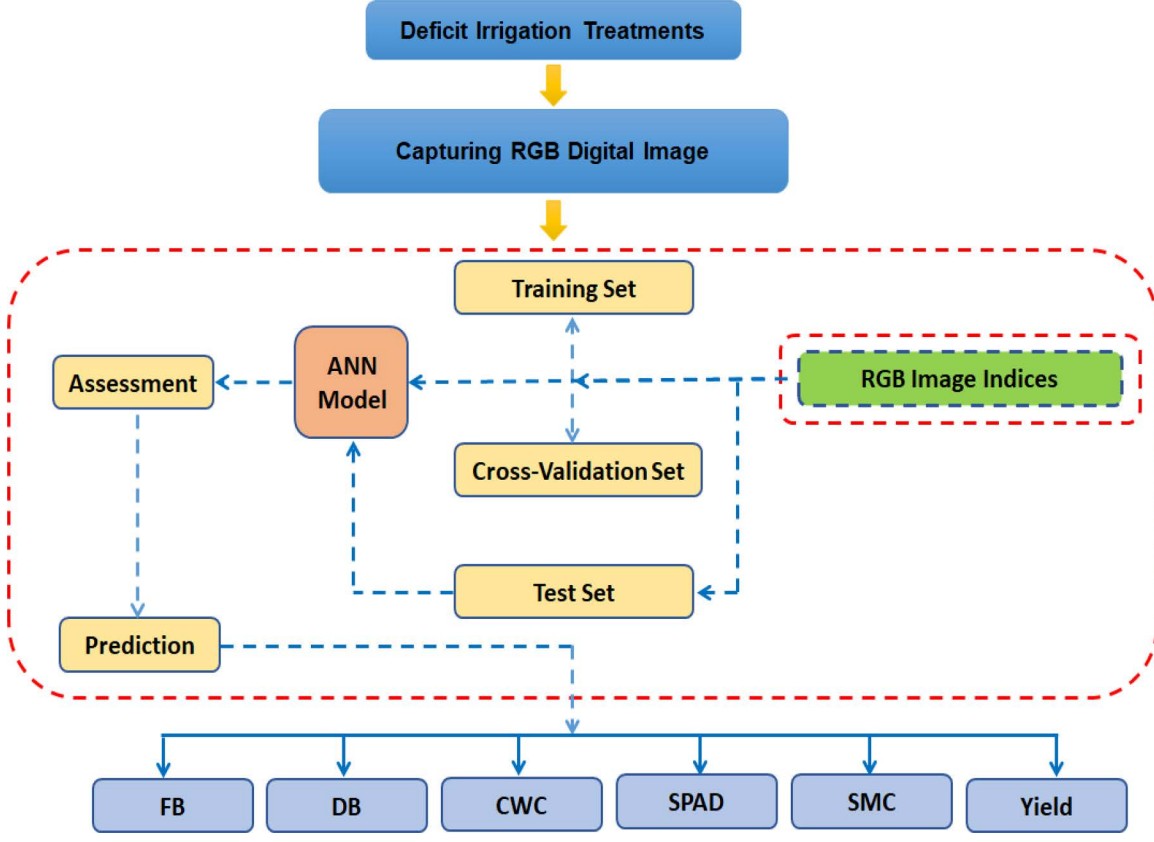

**Fig 2. Schematic diagram for predicting various plant traits using RGB image indices with ANN models.**

**Table 2. Types of activation function in ANN models.**

| Name | Equations |
| --- | --- |
| Hyperbolic Tangent (Tanh) | $f(x) = \frac{(e^x - e^{-x})}{(e^x + e^{-x})}$ |
| Logistic (Sigmoid) | $f(x) = \frac{1}{1 + e^{-x}}$ |
| Rectified Linear Unit (ReLU) | $f(x) = \max(0, x)$ |
| Linear (Identity) | $f(x) = x$ |

To prevent overfitting and guide hyperparameter selection, a 5-fold cross-validation procedure was applied exclusively to the training set. The best-performing model with the optimal combination of hyper-parameters and input feature was selected based on the lowest root mean squared error (RMSE) and the highest coefficient of determination ($R^2$) value, indicating a better fit to the data.

**2.5.4. Model evaluation.** Models performance were evaluated by comparing the predicted values against the observed values during training and testing sets using metrics such as determination coefficient ($R^2$), root mean squared error (RMSE) value (refer to Eqs (6) and (7)) and normalized root mean squared error (NRMSE) value (refer to Eq (8)) to assess the accuracy and reliability of the predictions [19,62]. The NRMSE, calculated as described by Wellens *et al.* [63], provides a quantitative measure (expressed as a percentage) of the relative difference between predicted and actual

values. According to Gholipoor and Nadali [23], NRMSE (%) values between 0 and 10% are regarded as "excellent," those between 10 and 20% as "good," values between 20 and 30% as "fair," and values over 30% as "poor."

$$R^2 = 1 - \frac{\sum (Y_a - Y_p)^2}{\sum (Y_a - \mu)^2} \tag{6}$$

$$RMSE = \sqrt{\frac{1}{N} \sum_{i=1}^{N} (Y_a - Y_p)^2} \tag{7}$$

$$NRMSE = RMSE * \frac{100}{\mu} \tag{8}$$

$Y_a$: represents the actual value that was calculated in the laboratory. $Y_p$: represents the predict value, $\mu$: represents the mean value, and N is the total number of data points.

## 2.6. Statistics analysis

The experiment was designed as a randomized complete block design (RCBD) with four replicates. All collected data were subjected to analysis of variance (ANOVA) to evaluate the response of plant trait measurements across both stages during two seasons under different irrigation levels. The statistical software package used for data analysis was SPSS version 28.0. To determine significant differences between means, Tukey post-hoc test was employed at significance levels of $P \leq 0.05$, 0.01, and 0.001. To examine the relationships between plant traits and RGB image indices, as well as between actual values and predicted values by ANN models, regression models were fitted. These regression analyses were conducted using Excel 2016 (v14.0). The significance of the relationships was assessed by calculating $R^2$ values, with a significance level set at 0.05, 0.01, and 0.001 probability levels for all relationships.

## 3. Results and discussion

### 3.1. Impact of irrigation levels on plant traits and yield of tomato

The combined analysis of variance conducted on different plant traits (Table 3) for tomato crops indicated that there were no significant differences observed between the growing seasons. Furthermore, the two-way interaction between season and irrigation regimes, as well as season and both stages, did not have a significant impact on the plant traits. Additionally, the three-way interaction among season, irrigation regimes, and both stages did not show any significant effects on the plant traits studied. However, highly significant differences ($P \leq 0.001$) were observed between irrigation regimes and both stages for all plant traits. However, the flowering stage and fruit ripening stage, as well as the interaction of irrigation regimes and the flowering stage or irrigation regimes and fruit ripening stage, were non-significant for all plant traits. Regarding all plant traits, it was non-significant, except for SPAD, which revealed highly significant differences. Finally, the ANOVA results for SMC show that all factors and interactions, including season (S), both Stages (BS), irrigation (I), S x BS, S * I, BS* I, and S * BR * I, were highly significant.

A one-way ANOVA revealed a statistically-significant difference in mean fresh biomass (FB) weights according to deficit irrigation regimes ($F(2)= 1027.48$, $p < 0.001$, $\eta^2 = 0.976$) and (($F(2)= 687.44$, $p < 0.001$, $\eta^2 = 0.964$) during flowering stage (FS) and fruit ripening stage (FRS), respectively. A Tukey post-hoc test revealed significant pairwise differences between treatments ($p < 0.001$) during FS and FRS, as shown in Fig 3a. The highest FB weights were observed with T100, followed by

**Table 3. Interaction effect of different irrigation levels, stage, and season for tomato crop.**

| | FB, ton/ha | DB, ton/ha | CWC, % | SPAD | SMC, % | Yield, ton/ha |
|---|---|---|---|---|---|---|
| Season (S) | NS | NS | NS | NS | *** | NS |
| Both Stages (BS) | *** | *** | *** | *** | *** | – |
| Irrigation (I) | *** | *** | *** | *** | *** | *** |
| S x BS | NS | NS | NS | NS | *** | – |
| S * I | NS | NS | NS | NS | *** | – |
| BS * I | NS | NS | NS | *** | *** | – |
| S * BS * I | NS | NS | NS | NS | *** | – |
| Flowering Stage (FS) | NS | NS | NS | NS | *** | – |
| FS*I | NS | NS | NS | NS | * | – |
| Fruit Ripening Stage (FRS) | NS | NS | NS | NS | *** | – |
| FRS*I | NS | NS | NS | NS | *** | – |

NS, *, and *** indicate non-significant, significance level of P ≤ 0.05, and significance level of P ≤ 0.001, respectively.

T75. The lowest values were recorded with T50 during the FS and FRS, as shown in Fig 3a. The FB weights decreased by 26% and 29.95% in the T75 and by 60.58% and 63.38% in the T50, compared to the T100 during the FS and FRS, respectively. Similarly, a one-way ANOVA revealed a statistically-significant difference in mean dry biomass (DB) weights according to different irrigation regimes ($F_{(2)}$= 484.73, $p < 0.001$, $\eta^2 = 0.950$) and (($F_{(2)}$= 297.20, $p < 0.0001$, $\eta^2 = 0.921$) during FS and FRS, respectively. A Tukey post-hoc test revealed significant pairwise differences between treatments ($p < 0.001$) during FS and FRS, as shown in Fig 3b. The DB decreased by 18.23% and 22.31% in the T75 and by 46.13% and 50.56% in the T50, compared to the T100 during the FS and FRS, respectively, as shown in Fig 3b. The observed reductions in DB and FB weights can be attributed to the adverse effects of water stress, which hindered photosynthetic processes due to decreased plant water and chlorophyll content. These findings align closely with the results reported by [6,8,64–66], which similarly reported that the full irrigation regimes yielded the highest DB and FB weights for tomato crops when compared to other deficit irrigation approaches. A one-way ANOVA revealed a statistically-significant difference in mean CWC according to different irrigation regimes ($F_{(2)}$= 540.14, $p < 0.001$, $\eta^2 = 0.955$) and (($F_{(2)}$= 476.10, $p < 0.001$, $\eta^2 = 0.949$) during FS and FRS, respectively. A Tukey post-hoc test revealed significant pairwise differences between treatments ($p < 0.001$) during FS and FRS, as shown in Fig 3c. The highest CWC values (87.64 and 86.8%) were observed at the T100, while the lowest values (82.95 and 82.57%) were observed at the T50. The T75 showed a slight reduction (86.46 and 85.82%) compared to the T100, respectively, as shown in Fig 3c. The results revealed a consistent pattern of reduced CWC with a decrease in the amount of irrigation water applied. So, scientists have used CWC to indicate plant water stress and schedule irrigation. A one-way ANOVA revealed a statistically-significant difference in mean SMC according to different irrigation regimes ($F_{(2)}$= 23.43, $p < 0.001$, $\eta^2 = 0.479$) and (($F_{(2)}$= 86.85, $p < 0.001$, $\eta^2 = 0.773$) during FS and FRS, respectively. A Tukey post-hoc test revealed significant pairwise differences between treatments ($p < 0.001$) during FS and FRS, as shown in Fig 3d. During the FS and FRS, the highest SMC values (24.78 and 22.81%) were observed at the T100, followed by T75 (20.61 and 18.30%). The lowest SMC values (16.44 and 13.79%) were achieved at the T50, respectively, as shown in Fig 3d. These observed are consistent with the findings reported by [67–69]. A one-way ANOVA revealed a statistically-significant difference in mean SPAD according to deficit irrigation regimes ($F_{(2)}$= 337.13, $p < 0.001$, $\eta^2 = 0.930$) and (($F_{(2)}$= 238.48, $p < 0.001$, $\eta^2 = 0.903$) during FS and FRS, respectively. A Tukey post-hoc test revealed significant pairwise differences between treatments ($p < 0.001$) during FS and FRS, as shown in Fig 3e. During the FS, the highest SPAD values, indicative of relative chlorophyll content, were observed at T75 (56.00), followed by T100 (53.87), and then T50 (46.27), as shown

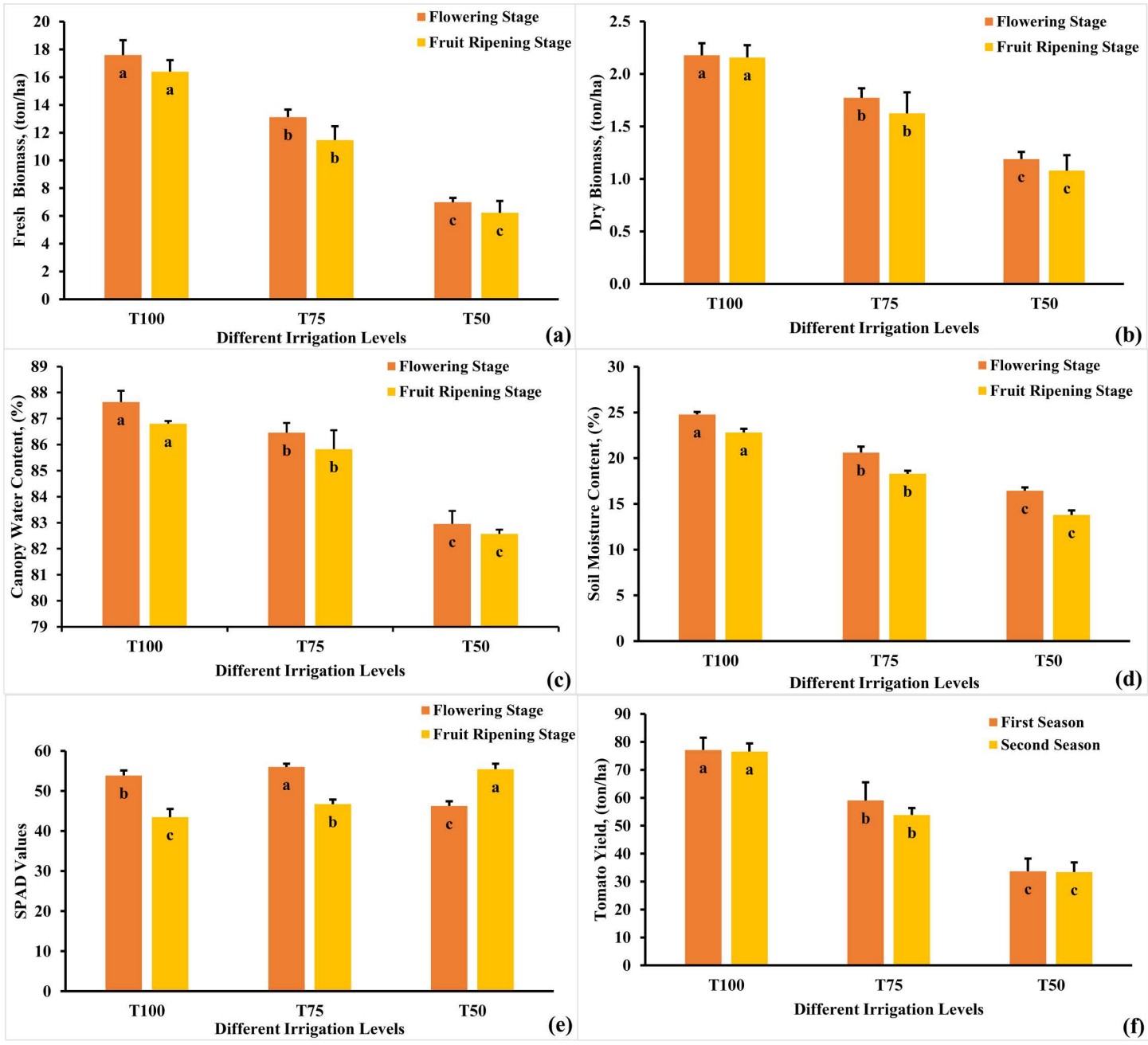

**Fig 3. Variation in the FB, DB, CWC, SMC, SPAD and yield of tomato crop in both seasons under different irrigation regime.** Means having the different alphabetical letter (s) are significantly differ at 0.001 level according to Tukey post-hoc test. Different alphabetical letter (a, b, c) indicates significant differences between treatments (e.g., T100 vs. T75 vs. T50) within the same growth stage.

in Fig 3e. The impact of water stress on chlorophyll, the primary pigment crucial for photosynthesis, was apparent. Plants under stress exhibited decreased chlorophyll content due to severely reduced CWC, leading to lower SPAD values, as seen in the case of T50. Conversely, a slight reduction in CWC could concentrate chlorophyll content in specific leaf areas, resulting in higher SPAD values, exemplified by T75. These results closely resemble the findings of [70,71]. For the FRS,

the highest SPAD values (55.45) were observed at T50, followed by T75 (46.71), and T100 (43.46) in both seasons, as shown in Fig 3e, consistent with the results reported by [66]. These results emphasized that SPAD values rise with increasing water stress during FRS. Conversely, SPAD values decreased in the full irrigation treatment due to its promotion of intensive photosynthesis, ultimately enhancing tomato yield.

A one-way ANOVA revealed that there was a statistically significant difference in mean tomato yield according to deficit irrigation regimes (F(2) = 464.252, p<0.001). The effect size, eta squared ($\eta^2$), was 0.948, indicating a large effect. Tukey post-hoc test revealed significant pairwise differences between treatments (p<0.001) during first and second seasons, as shown in Fig 3f. The results also demonstrated that the highest tomato yield was achieved under T100 (77.09 and 76.56 tons/ha) during the first and second seasons, respectively, followed by T75, which yielded 59.02 and 53.85 ton/ha. The lowest tomato yield was recorded under T50, with 33.69 and 33.40 ton/ha for both seasons, respectively, as shown in Fig 3f. These findings align with previous studies conducted by [6,8,72]. Lower CWC signifies water stress, leading to decreased photosynthetic activity and poor plant health. Reduced chlorophyll (SPAD values) limits photosynthesis efficiency, resulting in stunted growth and fewer flowers and fruits, which are crucial for yield. Additionally, lower FB and DB weights indicates diminished overall health and nutrient accumulation, correlating with reduced yield, as plants with insufficient biomass struggle to support fruit development and maturation. Therefore, accurately detecting plant variables is crucial for optimizing crop yield.

Tomatoes are the most popular vegetables grown in different parts of the world. However, because of its shallow root systems, it is vulnerable to soil moisture stress. In tomato plants, the majority of the root density is localized in the soil between 0 and 15 centimeters deep [64]. Thus, to achieve higher returns and water productivity in tomato production in open field conditions, irrigation management and maintaining appropriate soil moisture conditions are essential. The CROPWAT program determined the irrigation scheduling for tomato crop. Before initiating the irrigation regimes, equal amounts of water were added to each treatment during the first 15 days (78.9 mm in the first season and 108.1 mm in the second season). During the first season, total gross irrigation depths was recorded 435.7 mm, 614.1 mm, and 792.5 mm, which corresponded to T50, T75, and T100, respectively, with a frequency of 31 irrigation events. In the second season, the total gross irrigation depths reached 513.2 mm, 715.75 mm, and 918.3 mm for T50, T75, and T100, respectively, accompanied by 37 irrigation events.

While the T100 treatment resulted in the highest yield, this study demonstrates that the T75 regime may offer a more favorable balance between crop performance and resource use. Although the T75 treatment led to a statistically significant reduction in yield compared to T100, it still produced a commercially viable yield (59.02 and 53.85 ton/ha) while achieving a substantial saving of 22.29% in irrigation water applied. Previous studies have reported a wide range of optimum irrigation needs for achieving high tomato yields, varying from 532 to 905 mm under hot and dry climate conditions [73,74]. The study findings revealed that the optimal irrigation depths for maximizing tomato yields differed significantly between the first (792.5 mm) and second (918.3 mm) growing seasons. These results emphasize the substantial variability in irrigation requirements for achieving optimal tomato yields, which can be attributed to various factors such as soil characteristics, irrigation methods, climate, and other management and environmental factors. The significant variation in irrigation needs observed among different studies further underscores the critical role of effective irrigation water management in enhancing crop productivity, particularly under conditions of water scarcity or deficit irrigation stress.

### 3.2. The RGB image indices for evaluating the plant traits

The combined analysis of variance conducted on different RGB image indices for tomato crops (Table 4) indicated that there were no significant differences observed between the growing seasons. Furthermore, the two-way interaction between season and irrigation regimes, as well as season and both stages, did not have a significant impact on the RGB image indices. Additionally, the three-way interaction among season, irrigation regimes, and both stages did not show any significant effects on the RGB image indices. However, highly significant differences (P ≤ 0.001) were observed between irrigation regimes and both stages for all RGB image indices, except for RB, which did not exhibit significant differences between irrigation regimes. However, the FS and FRS, as well as the interaction of irrigation regimes and the FS or

**Table 4. Combined analysis of variance for RGB image indices among three irrigation levels for tomato crop for both stages during both seasons.**

| Treatment | R% | G% | B% | GR | RB | GB | MGVRI | RGVBI | ExR | ExG | ExGR | VEG | CIVE | COM | VARI | GRVI | NDI | TGI | IPCA | GLI | MExG |
|---|---|---|---|---|---|---|---|---|---|---|---|---|---|---|---|---|---|---|---|---|---|
| Season (S) | NS | NS | NS | NS | NS | NS | NS | NS | NS | NS | NS | NS | NS | NS | NS | NS | NS | NS | NS | NS | NS |
| Both Reading (BR) | *** | *** | *** | *** | * | *** | *** | *** | *** | *** | *** | *** | *** | *** | *** | *** | *** | *** | *** | *** | *** |
| Treatment (T) | *** | *** | *** | *** | NS` | *** | *** | *** | *** | *** | *** | *** | *** | *** | *** | *** | *** | *** | *** | *** | *** |
| S x BR | NS | NS | NS | NS | NS | NS | NS | NS | NS | NS | NS | NS | NS | NS | NS | NS | NS | NS | NS | NS | NS |
| S * T | NS | NS | NS | NS | NS | NS | NS | NS | NS | NS | NS | NS | NS | NS | NS | NS | NS | NS | NS | NS | NS |
| BR * T | NS | *** | NS | *** | NS | *** | ** | *** | NS | * | NS | *** | NS | NS | NS | NS | NS | * | NS | NS | NS |
| S * BR * T | NS | NS | NS | NS | NS | NS | NS | NS | NS | NS | NS | NS | NS | NS | NS | NS | NS | NS | NS | NS | NS |
| First Reading (FR) | NS | NS | NS | NS | NS | NS | NS | NS | NS | NS | NS | NS | NS | NS | NS | NS | NS | NS | NS | NS | NS |
| FR*T | NS | NS | NS | NS | NS | NS | NS | NS | NS | NS | NS | NS | NS | NS | NS | NS | NS | NS | NS | NS | NS |
| Second Reading (SR) | NS | NS | NS | NS | NS | NS | NS | NS | NS | NS | NS | NS | NS | NS | NS | NS | NS | NS | NS | NS | NS |
| SR*T | NS | NS | NS | NS | NS | NS | NS | NS | NS | NS | NS | NS | NS | NS | NS | NS | NS | NS | NS | NS | NS |

NS, *, **, and *** indicate non-significant, significance level of $P \leq 0.05$, significance level of $P \leq 0.01$, and significance level of $P \leq 0.001$, respectively. The complete names of the abbreviated RGB image indices can be found in Table 1.

irrigation regimes and FRS, were non-significant for all RGB image indices. Additionally, the two-way interaction of both stages and irrigation regimes were non-significant for all RGB image indices, except for the G (%), GR, GB, MGVRI, RGVBI, ExG, VEG, and TGI indices. Regarding all plant traits, it was non-significant, except for SPAD, which revealed highly significant differences. Finally, the ANOVA results for SMC show that all factors and interactions, including season (S), both Stages (BS), irrigation (I), S x BS, S * I, BS* I, and S * BR * I, were highly significant.

The average RGB image indices of tomato plants that were subjected to different irrigation regimes are presented in Table 5. The RGB image indices in the three irrigation treatments showed similar trends during both growth stages. The mean values for RGB image indices were lower in the green band at the fruit ripening stage and higher in the red and blue bands compared to the flowering stage. This is due to increased water stress sensitivity and declining chlorophyll content during senescence. Our findings concur with those of [75], who reported that when bean plants flower, the vegetation index values decrease. The RGB image indices' mean values differed based on irrigation regimes, indicating that deficit irrigation had an impact on the percentages of the red (R%), green (G%), and blue (B%) channels. In comparison to the fully irrigated plants, the water-stressed plants had higher R% and B% and lower G%. According to Vinod et al. [76], this indicates that fully irrigated plants absorb more visible light (the red and blue bands) for photosynthetic processes. The concentration of leaf photosynthetic pigment was found to be impacted by water stress, as seen by the stressed treatments recording the lowest value at G%. This result implies that the tomato crops' color attributes were significantly impacted by the different irrigation regimes. Consequently, this highlights the viability and cost-effectiveness of utilizing the RGB method as a practical monitoring tool for effectively managing deficit irrigation. Numerous studies have shown a strong correlation between the visible light spectrum (400–700 nm), canopy water, and chlorophyll content [77,78]. In addition, Qian et al. [79] found a negative correlation between SPAD and the blue and red channels, while Mercado-Luna et al. [80] found a positive correlation between SPAD and the green channel. It seems that certain plant features related to development and water status could be efficiently monitored using the three visible light channels (RGB).

In this study, the various vegetation indices obtained from RGB images could also be used to assess FB, DB, CWC, SPAD, SMC, and tomato yield. The various RGB image indices obtained by the digital camera showed notable differences between the three irrigation treatments, as can be observed in Table 5. Significantly lower values were obtained under

**Table 5. Comparison the mean values for RGB image indices among three different irrigation levels for tomato crop during both stages and both Seasons.**

| | Treat. | R % | G % | B % | GR | RB | GB | MGVRI | RGVBI | ExR | ExG |
|---|---|---|---|---|---|---|---|---|---|---|---|
| R1 (Both Seasons) | T100 | 0.23±0.03c | 0.57±0.04a | 0.20±0.02c | 2.59±0.63a | 1.15±0.19a | 2.88±0.36a | 0.71±0.11a | -0.12±0.04a | -48.49±16.23a | 139.08±18.45a |
| | T75 | 0.30±0.03b | 0.46±0.02b | 0.25±0.03b | 1.59±0.22b | 1.23±0.30a | 1.92±0.32b | 0.42±0.10b | -0.24±0.03b | -9.02±9.54b | 70.54±7.69b |
| | T50 | 0.36±0.06a | 0.31±0.03c | 0.33±0.05a | 0.88±0.21c | 1.14±0.30a | 0.95±0.17c | -0.14±0.23c | -0.54±0.07c | 32.43±16.50a | -11.83±15.75c |
| R2 (Both Seasons) | T100 | 0.28±0.06c | 0.42±0.02a | 0.30±0.04c | 1.58±0.49a | 0.99±0.29a | 1.45±0.15a | 0.37±0.20a | -0.31±0.05a | -4.31±19.16a | 54.30±10.78a |
| | T75 | 0.36±0.08b | 0.31±0.02b | 0.34±0.03b | 0.92±0.19b | 1.04±0.21a | 0.92±0.05b | -0.10±0.19b | -0.54±0.05b | 38.07±18.94b | -14.39±13.13b |
| | T50 | 0.43±0.04a | 0.18±0.03c | 0.41±0.04a | 0.44±0.09c | 1.01±0.22a | 0.44±0.10c | -0.67±0.10c | -0.83±0.05c | 74.58±15.07c | -88.41±18.30c |

| | Treat | ExGR | VEG | CIVE | COM | VARI | GRVI | NDI | TGI | IPCA | GLI | MExG |
|---|---|---|---|---|---|---|---|---|---|---|---|---|
| R1 (Both Seasons) | T100 | 187.57±34.38a | 2.68±0.52a | -36.81±7.7c | 79.22±12.42a | 0.34±0.08a | 0.43±0.10a | 182.80±12.48a | 70.19±8.61a | 64.34±5.83a | 0.45±0.07a | 88.91±12.94a |
| | T75 | 79.56±15.29b | 1.68±0.15b | -8.82±3.19b | 38.80±5.33b | 0.17±0.04b | 0.22±0.06b | 156.59±7.90b | 36.26±4.03b | 35.74±6.68b | 0.26±0.04b | 45.29±5.60b |
| | T50 | -44.26±29.93c | 0.90±0.15c | 24.59±6.56a | -8.01±10.61c | -0.05±0.08c | -0.07±0.12c | 118.52±15.52c | -5.32±7.51c | -2.38±9.13c | -0.06±0.08c | -5.22±12.13c |
| R2 (Both Seasons) | T100 | 58.61±29.15a | 1.51±0.27a | -2.38±4.84c | 30.55±9.80a | 0.14±0.08a | 0.20±0.12a | 153.92±15.92a | 26.96±4.17a | 23.04±6.81a | 0.19±0.05a | 38.77±10.78a |
| | T75 | -52.46±31.81b | 0.92±0.12b | 25.78±5.79b | -10.72±10.9b | -0.04±0.06b | -0.05±0.1b | 121.46±12.53b | -6.94±5.18b | -5.16±3.80b | -0.05±0.1b | -4.83±11.78b |
| | T50 | -162.99±30.9c | 0.44±0.08c | 55.77±7.49a | -52.54±11.26c | -0.23±0.06c | -0.39±0.09c | 77.95±11.08c | -44.34±9.5c | -39.57±11.9c | -0.40±0.08c | -50.21±11.5c |

The means followed by the same letter are not significantly different from each other, as determined by Tukey's HSD post hoc test at a significance level of $P \leq 0.05$. The complete names of the abbreviated RGB image indices can be found in Table 1.

T50 than under T100. Furthermore, four RGB image indices (R%, B%, ExR and CIVE) showed an inverse proportionality with the plant features, while the majority of the collected indices showed a strong positive association. However, it is important to mention that the RB index failed to demonstrate any relationship with the plant traits. When examining the flowering stage data, $R^2$ values were observed ranging from 0.60 to 0.93 for FB, 0.63 to 0.92 for DB, 0.46 to 0.85 for CWC, 0.49 to 0.80 for SPAD, 0.27 to 0.49 for SMC, and 0.59 to 0.88 for tomato plant yield (Fig 4a). Moving on to the fruit ripening stage, $R^2$ values were ranged from 0.61 to 0.94 for FB, 0.57 to 0.91 for DB, 0.50 to 0.84 for CWC, 0.52 to 0.82 for SPAD, 0.44 to 0.74 for SMC, and 0.51 to 0.89 for tomato plant yield (Fig 4b). When the data was combined for both stages during both seasons, the $R^2$ values were ranged from 0.52 to 0.79 for FB, 0.49 to 0.76 for DB, 0.44 to 0.72 for CWC, 0.10 to 0.43 for SPAD, 0.35 to 0.59 for SMC, and 0.42 to 0.63 for tomato plant yield (Fig 4c). Similarly, Johansen et al. [81] highlighted the usefulness of these indices as a tool for predicting the fresh biomass and tomato yield per plant under various growing conditions by using vegetation indices based on red, green, and blue (RGB) photographs. Similar results for the crops of beans and cassava were recently published by research [82,83], indicating the usefulness of RGB image indices in measurements of plant attributes.

Our results of the image analysis showed that the blue and red channels alone is not suitable for predicting plant traits and yield estimation. The RB ratio does not take into account the green component (G) of vegetation, which is known to be important for assessing vegetation health and biomass, so this index may not capture all the relevant factors influencing the parameters of interest. Green vegetation reflects light differently compared to red and blue wavelengths. RGB image indices that incorporate the green component (G), such as the G%, ExG, and MExG, are sensitive to the amount of green vegetation in an image. Since greenness is related to chlorophyll content and overall plant health [84]. By leveraging these indices, farmers and researchers can gain deeper insights into plant health and growth dynamics. Accurate detection of various plant traits that are closely linked to soil moisture availability can have significant implications for determining irrigation thresholds, scheduling, and improving irrigation water management. It can also contribute to water conservation efforts. For example, studies by [85,86] have shown that deficit irrigation can lead to notable reductions in crop yield, which can be predicted by indirectly assessing total amount of dry matter accumulation during the early growth stage. Regular and simultaneous evaluation of these plant traits utilizing rapid, easy, and non-destructive techniques can therefore be extremely important in optimizing irrigation water usage while reaching target yields through efficient deficit irrigation management.

## 3.3. Modeling RGB image indices through artificial neural network

The artificial neural network model has showcased remarkable performance as a regression technique, particularly in pattern recognition and function determination. Compared to traditional methods. ANNs are capable of drawing conclusions, handling incomplete information, and being less affected by outliers [87,88]. By employing distinct RGB image indices as independent variables during neural network training, accurate anticipation of the examined plant stress indicators (dependent variables) and tomato yield was achieved. (Tables 6–8) present the optimal combination of RGB image indices, hyperparameters, and outputs from the ANN model in terms of $R^2$, RMSE, and NRMSE. This combination of RGB image indices was explored through training, cross-validation, and assessment of the test dataset, with the objective of evaluating plant traits such as FB, DB, CWC, SPAD, SMC, and tomato yield (TY). For most applications, the determination of hyperparameters must be made by the designer, as there is no definitive rule to determine the optimal number of hidden layers and neurons to solve a given problem [76,89].

### 3.3.1. Tomato models using flowering stage data.
Using the data acquired during the flowering stage for both seasons, some of the ANN models were able to predict plant traits and tomato yield (TY) with a high level of accuracy (Table 6). The model's performance during testing phase using flowering stage data is depicted in Fig 5. The training output showed that, among the different ANN models, the ANNT-FB1 model was found to be more accurate in predicting the FB, with a $R^2$ of 0.99 and an RMSE of 22.30 g/plant. Based on the NRMSE value (3.43%), this indicates that this

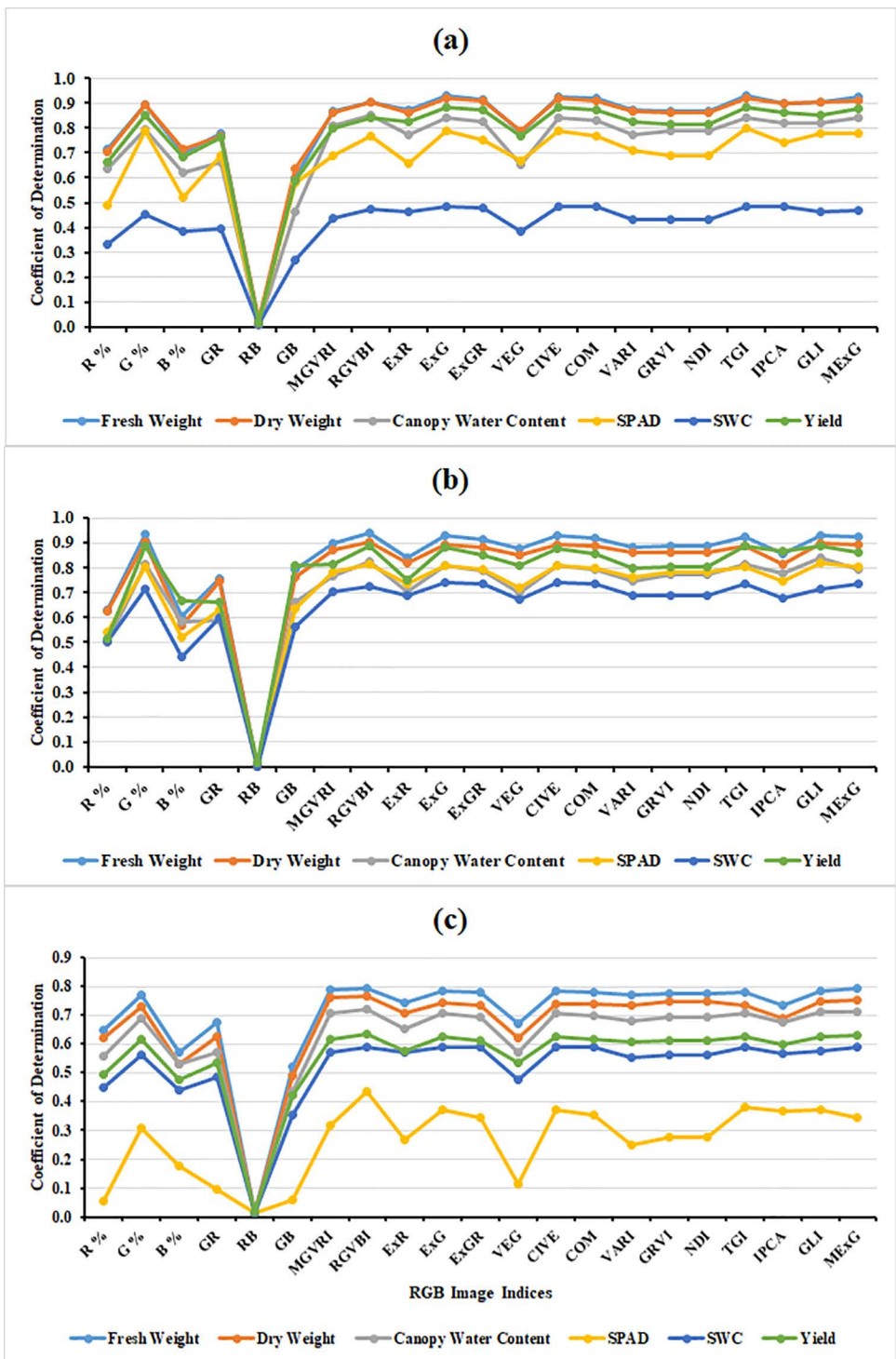

**Fig 4. Displays the coefficient of determination values indicating the strength of the relationships between the RGB image indices and plant traits, as well as tomato yield, under varying irrigation levels during both seasons for (a) the flowering stage data, (b) the fruit ripening stage data, and (c) the data combined for both stages.** The complete names of the abbreviated RGB image indices can be found in Table 1.

**Table 6. Performance of different ANN models for prediction of plant traits and tomato yield after training and testing using different RGB image indices at flowering stage.**

| Model | Optimal Features | Hyper-parameter (Z, L, N, I, O) | Training | | | Testing | | |
|---|---|---|---|---|---|---|---|---|
| | | | $R^2$ | RMSE (gm/plant) | NRMSE (%) | $R^2$ | RMSE (gm/plant) | NRMSE (%) |
| ANNT-FB1 | R %, G %, B %, GR, RB, GB, MGVRI, RGVBI, ExR, ExG, ExGR, VEG, CIVE, COM, VARI, GRVI, NDI, TGI, IPCA, GLI, MExG | (tanh, 1, 7, 800, QN) | 0.99*** | 22.30 | 3.43 | 0.95*** | 47.58 | 7.32 |
| ANNT-DB1 | R %, G %, B %, GR, GB, MGVRI, RGVBI, ExR, ExG, ExGR, VEG, CIVE, COM, VARI, GRVI, NDI, TGI, IPCA, GLI, MExG | (logistic, 1, 5, 600, QN) | 0.97*** | 3.33 | 5.53 | 0.92*** | 5.68 | 9.52 |
| ANNT -CWC1 | R %, G %, B %, GR, GB, MGVRI, RGVBI, ExR, ExG, ExGR, VEG, CIVE, COM, VARI, GRVI, NDI, TGI, IPCA, GLI, MExG | (tanh, 1, 8, 500, QN) | 0.97*** | 0.34 | 5.69 | 0.97*** | 0.37 | 6.40 |
| ANNT-SPAD1 | R %, G %, MGVRI, RGVBI, ExR, ExG, ExGR, CIVE, COM, VARI, GRVI, NDI, TGI, IPCA, GLI, MExG | (logistic, 1, 9, 500, QN) | 0.98*** | 0.70 | 5.56 | 0.87*** | 1.47 | 12.07 |
| ANNT-SMC1 | R %, G %, B %, GR, GB, MGVRI, RGVBI, ExR, ExG, ExGR, VEG, CIVE, COM, VARI, GRVI, NDI, TGI, IPCA, GLI, MExG | (relu, 1, 4, 600, QN) | 0.78*** | 2.23 | 13.81 | 0.71*** | 2.53 | 16.05 |
| ANNT-TY1 | R %, G %, B %, GR, GB, MGVRI, RGVBI, ExR, ExG, ExGR, VEG, CIVE, COM, VARI, GRVI, NDI, TGI, IPCA, GLI, MExG | (relu, 1, 9, 500, QN) | 0.95*** | 0.19 | 7.64 | 0.86*** | 0.30 | 12.18 |

"Z" indicates the activation function, "L" denotes the layer number, "N" denotes the neuron number for each hidden layer, "I" indicates the number of iterations, and "O" signifies the optimization method. *** indicate significance level of $P \leq 0.001$.

model's prediction accuracy falls within the excellent range (0–10%). The ANNT-DB1 model was found to be more accurate in predicting the DB with a $R^2$ of 0.97 and a RMSE of 3.33 g/plant. Based on the NRMSE value (5.53%), the model's prediction accuracy falls within the excellent prediction (0–10%) range. Considering the prediction accuracy indicator (NRMSE), the models ANNT-CWC1, ANNT-SPAD1, and ANNT-TY1 come under the excellent prediction class (0–10%). On the other hand, the ANNT-SMC1 model is in the 10–20% good prediction class. Table 6 displays also the testing results of various ANN models for the prediction of tomato yield and plant traits using the flowering data set. The ANNT-FB1, ANNT-DB1, and ANNT-CWC1 models are classified under the excellent prediction class (0–10%) based on the model prediction error indicator (NRMSE). On the other hand, the models ANNT-SPAD1, ANNT-SMC1, and ANNT-TY1 fall into the good prediction class (10–20%). These models' $R^2$ values varied from 0.71 to 0.97.

**3.3.2. Tomato models using fruit ripening stage data.** The models output to simulate plant traits and tomato yield (TY) using fruit ripening stage data is shown in Table 7. The model's performance during testing phase using fruit ripening stage data is depicted in Fig 6. The training output showed that among the different ANN models, the ANNT-FB2 model was found to be more accurate in predicting the FB, with a $R^2$ of 0.98 and an RMSE of 26.50 g/plant. Based on the NRMSE value (4.51%), this indicates that this model's prediction accuracy falls under the category of excellent prediction (0–10%). The ANNT-DB2 model was found to be more accurate in predicting the DB, with a R2 of 0.98 and a RMSE of 3.48 g/plant. Based on the NRMSE value (5.23%), the model's prediction accuracy falls under the category of excellent prediction (0–10%). Considering the prediction accuracy indicator (NRMSE), the models ANNT-CWC2 and ANNT-TY2 come under the excellent prediction class (0–10%), and the ANNT-SPAD2 and ANNT-SMC2 models fall within the good prediction class (10–20%). The testing results for various ANN models in terms of the prediction of plant traits and tomato yield are shown in Table 7. Based on NRMSE, the ANNT-FB2 (8.88%) and ANNT-TY2 (7.73%) models are classified under the excellent prediction class (0–10%). The $R^2$ of these models is 0.94 to 0.94, respectively. Further, the

**Table 7. Performance of different ANN models for prediction of plant traits and tomato yield after training and testing using different RGB image indices at fruit ripening stage.**

| Model | Optimal Features | Hyper-parameter (Z, L, N, I, O) | Training | | | Testing | | |
|---|---|---|---|---|---|---|---|---|
| | | | R² | RMSE (gm/plant) | NRMSE (%) | R² | RMSE (gm/plant) | NRMSE (%) |
| ANNT-FB1 | G %, MGVRI, RGVBI, ExR, ExG, ExGR, VEG, CIVE, COM, VARI, GRVI, NDI, TGI, IPCA, GLI, MExG | (logistic, 1, 9, 500, QN) | 0.98*** | 26.50 | 4.51 | 0.94*** | 50.81 | 8.88 |
| ANNT-DB1 | R %, G %, B %, GR, RB, GB, MGVRI, RGVBI, ExR, ExG, ExGR, VEG, CIVE, COM, VARI, GRVI, NDI, TGI, IPCA, GLI, MExG | (logistic, 1, 7, 700, QN) | 0.98*** | 3.48 | 5.23 | 0.88*** | 7.97 | 12.48 |
| ANNT-CWC1 | G %, MGVRI, RGVBI, ExG, ExGR, CIVE, COM, VARI, GRVI, NDI, TGI, IPCA, GLI, MExG | (tanh, 1, 7, 500, QN) | 0.97*** | 0.31 | 6.44 | 0.81*** | 0.79 | 17.04 |
| ANNT-SPAD1 | G %, GR, GB, MGVRI, RGVBI, ExR, ExG, ExGR, VEG, CIVE, COM, VARI, GRVI, NDI, TGI, IPCA, GLI, MExG | (tanh, 1, 7, 600, QN) | 0.90*** | 1.73 | 10.81 | 0.83*** | 1.92 | 12.72 |
| ANNT-SMC1 | R %, G %, GR, GB, MGVRI, RGVBI, ExR, ExG, ExGR, VEG, CIVE, COM, VARI, GRVI, NDI, TGI, IPCA, GLI, MExG | (tanh, 1, 7, 500, QN) | 0.81*** | 1.72 | 12.53 | 0.75*** | 2.27 | 16.67 |
| ANNT-TY1 | R %, G %, B %, GR, RB, GB, MGVRI, RGVBI, ExR, ExG, ExGR, VEG, CIVE, COM, VARI, GRVI, NDI, TGI, IPCA, GLI, MExG | (tanh, 1, 4, 500, QN) | 0.96*** | 0.16 | 6.27 | 0.94*** | 0.19 | 7.73 |

"Z" indicates the activation function, "L" denotes the layer number, "N" denotes the neuron number for each hidden layer, "I" indicates the number of iterations, and "O" signifies the optimization method. *** indicate significance level of P ≤ 0.001.

**Table 8. Performance of different ANN models for prediction of plant traits and tomato yield after training and testing using different RGB image indices at both stages.**

| Model | Optimal Features | Hyper-parameter (Z, L, N, I, O) | Training | | | Testing | | |
|---|---|---|---|---|---|---|---|---|
| | | | R² | RMSE (gm/plant) | NRMSE (%) | R² | RMSE (gm/plant) | NRMSE (%) |
| ANNT-FB1 | R %, G %, GR, MGVRI, RGVBI, ExR, ExG, ExGR, VEG, CIVE, COM, VARI, GRVI, NDI, TGI, IPCA, GLI, MExG | (relu, 1, 6, 500, QN) | 0.96*** | 41.73 | 5.89 | 0.84*** | 87.75 | 14.16 |
| ANNT-DB1 | R %, G %, GR, MGVRI, RGVBI, ExR, ExG, ExGR, VEG, CIVE, COM, VARI, GRVI, NDI, TGI, IPCA, GLI, MExG | (logistic, 1, 5, 800, QN) | 0.95*** | 4.89 | 7.07 | 0.90*** | 7.28 | 11.17 |
| ANNT-CWC1 | G %, RB, MGVRI, RGVBI, ExR, ExG, ExGR, CIVE, COM, VARI, GRVI, NDI, TGI, IPCA, GLI, MExG | (logistic, 1, 5, 500, QN) | 0.89*** | 0.66 | 10.86 | 0.80*** | 0.91 | 16.67 |
| ANNT-SPAD1 | G %, B %, GR, RB, GB, VEG | (tanh, 1, 5, 500, QN) | 0.67*** | 2.99 | 18.67 | 0.72*** | 2.71 | 17.12 |
| ANNT-SMC1 | R %, G %, B %, GR, MGVRI, RGVBI, ExR, ExG, ExGR, VEG, CIVE, COM, VARI, GRVI, NDI, TGI, IPCA, GLI, MExG | (logistic, 1, 9, 800, QN) | 0.70*** | 2.60 | 15.62 | 0.55*** | 3.28 | 20.53 |
| ANNT-TY1 | R %, G %, B %, GR, GB, MGVRI, RGVBI, ExR, ExG, ExGR, VEG, CIVE, COM, VARI, GRVI, NDI, TGI, IPCA, GLI, MExG | (tanh, 1, 9, 800, QN) | 0.92*** | 0.23 | 8.91 | 0.83*** | 0.37 | 14.27 |

"Z" indicates the activation function, "L" denotes the layer number, "N" denotes the neuron number for each hidden layer, "I" indicates the number of iterations, and "O" signifies the optimization method. *** indicate significance level of P ≤ 0.001.

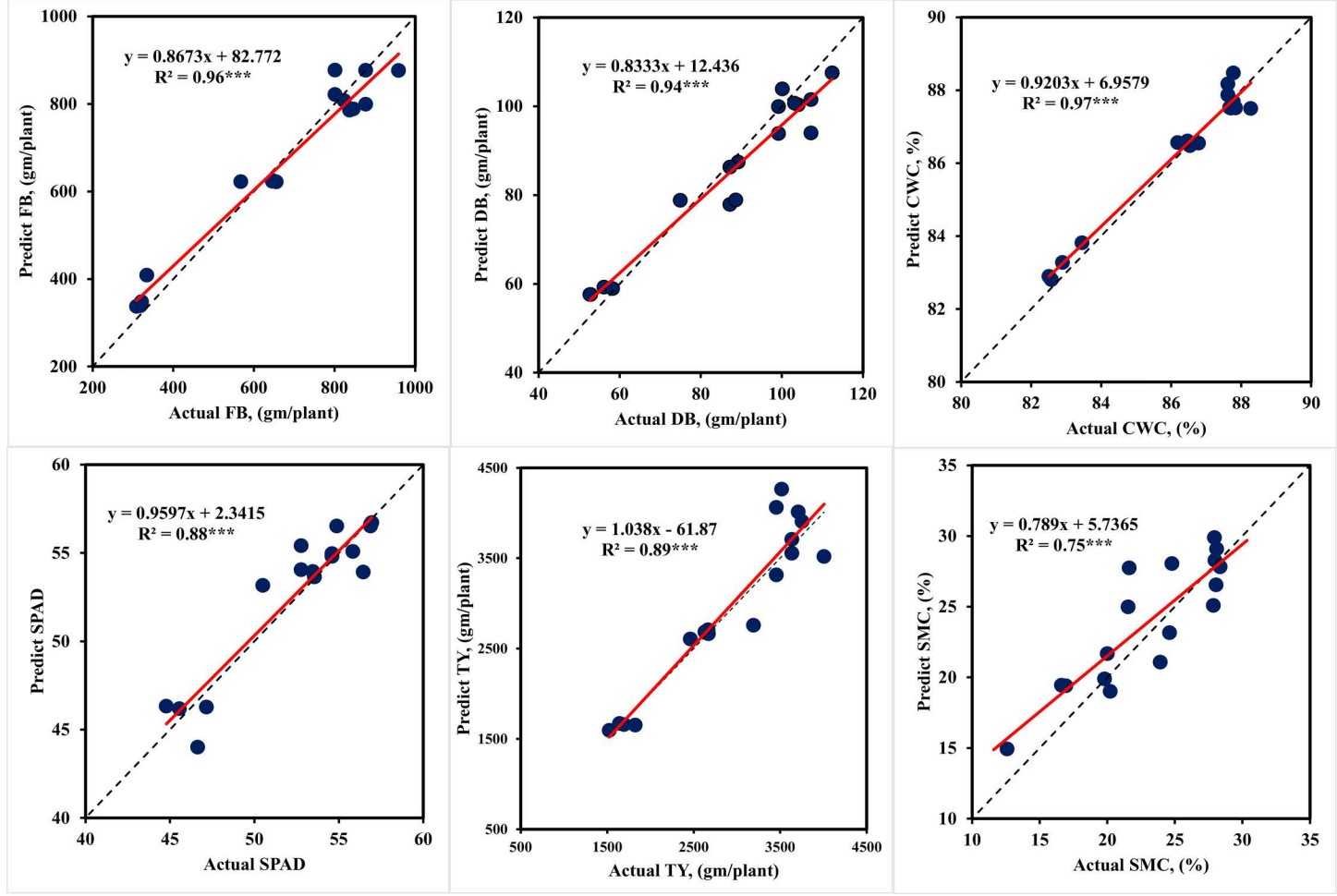

**Fig 5. Performances of ANN models during model testing to predict plant traits and tomato yield under different irrigation levels using flowering stage data testing.** *** indicate significance level of P ≤ 0.001.

ANNT-DB2, ANNT-CWC2, ANNT-SPAD2, and ANNT-SMC2 models are categorized under the good prediction class (10–20%). The $R^2$ of these models varied from 0.75 to 0.88.

### 3.3.3. Tomato models using both stages data.

Using the combined data for both stages during both seasons, ANN models were created with high accuracy to simulate plant traits and tomato yield (TY). The model training results for these models are shown in Table 8. The model's performance during testing phase is depicted in Fig 7. ANNT-FB3 model was found to be more accurate in predicting the FB, with a $R^2$ of 0.96 and an RMSE of 41.73 g/plant. Based on the NRMSE value (5.89%), the model's prediction accuracy falls under the excellent prediction category (0–10%). The ANNT-DB3 model was found to be highly accurate in DB prediction, with a $R^2$ of 0.95 and a RMSE of 4.89 g/plant. Based on the NRMSE value (7.07%), the model's prediction accuracy falls within the excellent prediction category (0–10%). Considering the NRMSE, the model ANNT-TY3 falls under the excellent prediction class (0–10%). ANNT-CWC3, ANNT-SPAD3, and ANNT-SMC3 models fall within the good prediction class (10–20%). The testing output of various ANN models in terms of the prediction of plant traits and tomato yield is shown in Table 8. Based on the NRMSE, the ANNT-FB3 (14.16%),

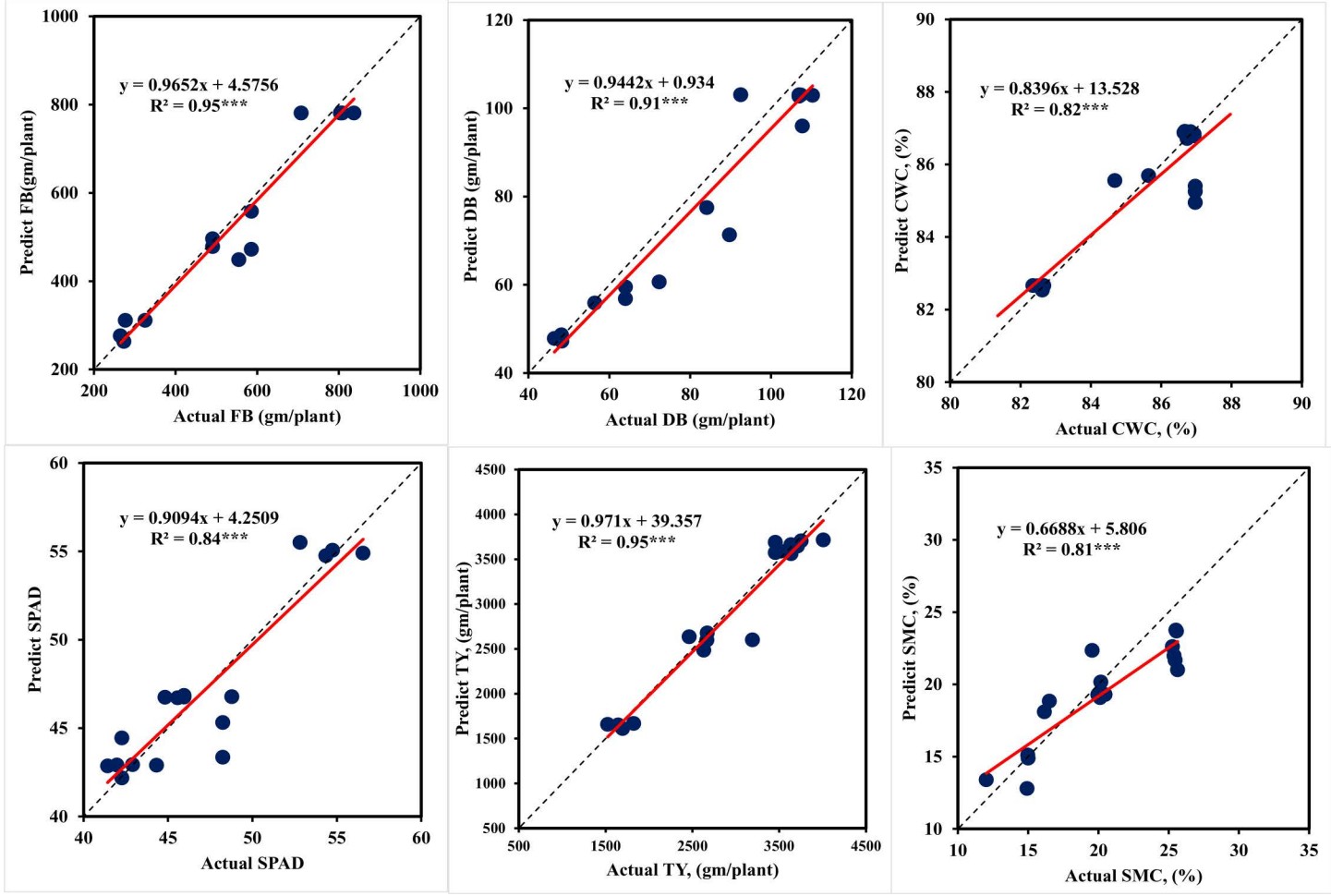

**Fig 6. Performances of ANN models during model testing to predict plant traits and tomato yield under different irrigation levels using fruit ripening stage data testing.** *** indicate significance level of P ≤ 0.001.

ANNT-DB3 (11.17%), ANNT-CWC3 (16.67%), ANNT-SPAD3 (17.12%), ANNT-SMC3 (20.53%), and ANNT-TY3 (14.27%) models are categorized under the good prediction class (10–10%). The $R^2$ of these models is 0.55 to 0.90, respectively.

The results in Tables 6–8 point to a possible overfit in the model. To ensure that our model did not learn unduly from the data, cross-validation was used to search for hyper-parameter optimization. To do cross-validation, the dataset was partitioned into k-folds, or random sets. One set chosen as the test set and the model was trained using the remaining sets. This process is repeated for each set used as the test set, and the average of the models is used to select the best model with the highest $R^2$ and lowest RMSE and NRMSE. The GridsearchCV library receives the machine learning model, hyper-parameters, and the chosen number of K-Folds. It then outputs the ideal estimator together with its optimal set of hyper-parameters. Seventy percent of the data was utilized as training data for cross validation, and the remaining thirty percent was used for testing. This division of the data was made according to a 70/30 ratio. The results of the ANN models are displayed in Tables 6–8. There is a small gap between the training and test outputs ($R^2$ and RMSE). For these values, overfitting is therefore less likely. This explains our decision to use K-Folds = 5. Figs 5–7 are

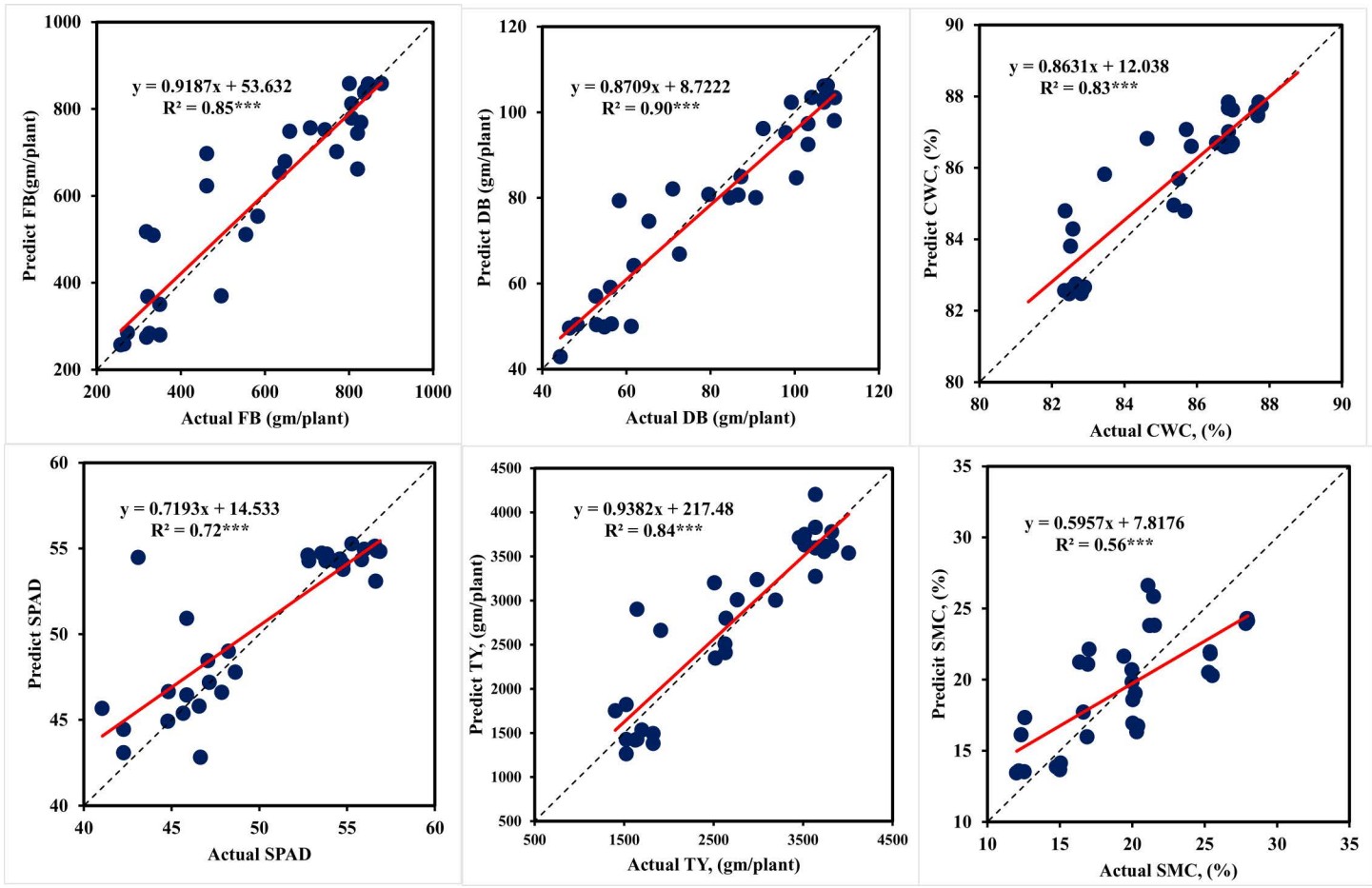

**Fig 7. Performances of ANN models during model testing to predict plant traits and tomato yield under different irrigation levels using both stages data testing.** *** indicate significance level of P ≤ 0.001.

presented that show the $R^2$ between the predicted and actual values. The linear regression positive slope is evident in the data representation. It was noted that some values differed somewhat from the large mass, potentially indicating bias. Overall, the data representation shows a linear relationship between the predicted and actual values, suggesting a reduced variance.

In this study, it is evident how powerful the response of RGB image indices of tomato in predicting plant traits and tomato yield under deficit irrigation when combined with an artificial neural network. Plant wilting and dehydration brought on by insufficient irrigation result in stomatal closure, changes in canopy water content and plant pigmentation. RGB image indices efficiently capture these changes [13]. ANN models were used to measure plant characteristics and tomato yield based on changes in these indices' values. Our results show good values for $R^2$, RMSE, and NRMSE. This may be attributed to the inclusion of strongly correlated RGB image indices that represent plant health. Even though forecast accuracy increases toward the harvest stage were noticed, the initial forecasts made during the flowering stage are more important for making decisions. Since there are no studies that compare to ours, we have instead contrasted our findings with other studies that employed artificial neural networks to predict plant growth and production. Hamdane et al. [90] integrated vegetation indices derived from RGB images with ANN and linear regression methods to evaluate the canopy

vigor and biomass of tomato, eggplant, and pepper plants. Their findings indicated that the ANN models outperformed the linear regression method in terms of predictive accuracy, despite a slight decrease in $R^2$ from 0.820 to 0.772 and a minor increase in NRMSE from 12.3% to 12.4%. Dutta Gupta and Pattanayak [91] employed ANN and linear models to analyze photometric features extracted from digital images of potato leaves for noninvasive estimation of chlorophyll content. The results favored the ANN model, showing superior performance with an $R^2$ of 0.90 compared to the linear model's $R^2$ of 0.41 in predicting potato leaf chlorophyll content. Hassanijalilian et al. [92] utilized RGB images of soybeans captured under field conditions with smartphones to develop an estimation ANN model for chlorophyll content, selecting indices with the strongest correlation to SPAD meter readings.

## 4. Conclusions

In conclusion, this study investigated the impact of water stress on tomato crop production and explored the utility of remote sensing and artificial neural network (ANN) models for assessing plant traits under different irrigation levels, growth stages, and growing seasons. The findings of the study provide valuable insights and conclusions. Firstly, the study confirmed the detrimental effects of limited water availability on tomato growth and productivity. It highlighted the importance of addressing water stress as a global challenge in agriculture. The study also explored the potential of RGB image indices as indirect indicators of plant traits related to water stress. Notably, several RGB image indices incorporating the green component (G), such as the G%, ExG, and MExG, exhibited strong positive relationship, with $R^2$ ranging between 0.52 and 0.94 for FB, 0.49 and 0.92 for DB, 0.44 and 0.85 for CWC, 0.10 and 0.82 for SPAD, 0.27 and 0.74 for SMC, and 0.42 and 0.89 for tomato yield. Conversely, indices like R%, B%, ExR and CIVE showed an inverse relationship with different plant traits. However, the red-blue simple ratio (RB) index, which does not consider the green component (G), did not show significant relationships with the plant traits. Furthermore, the study demonstrated the effectiveness of ANN models based on RGB image indices in predicting plant traits across irrigation levels, growth stages, and growing seasons. The high prediction accuracy achieved by the ANN models, underscores the reliability and practicality of this approach for managing tomato crop growth and production, as indicated by the $R^2$ values ranging from 0.84 to 0.99 for FB, 0.88 to 0.98 for DB, 0.81 to 0.97 for CWC, 0.67 to 0.98 for SPAD, 0.55 to 0.81 for SMC, and 0.83 to 0.96 for tomato yield. Overall, integrating strongly correlated RGB image indices that reflect plant health into ANN models can serve as a practical tool for effectively managing the growth and production of tomato crops under deficit irrigation conditions. The insights gained from this research can contribute to the development of precision farming practices and strategies to optimize crop yield and mitigate the impact of limited irrigation water availability on agricultural productivity.

## Author contributions

**Conceptualization:** Mohamed S. Abd El-baki, Mohamed Maher Ibrahim, Ali Salem, Nadia G. Abd El-Fattah.

**Data curation:** Mohamed S. Abd El-baki, Mohamed Maher Ibrahim, Ahmed Elbeltagi, Ali Salem.

**Formal analysis:** Mohamed S. Abd El-baki, Mohamed Maher Ibrahim, Salah Elsayed, Nadia G. Abd El-Fattah.

**Funding acquisition:** Ali Salem.

**Investigation:** Mohamed S. Abd El-baki, Salah Elsayed, Ahmed Elbeltagi, Ali Salem.

**Methodology:** Mohamed S. Abd El-baki, Salah Elsayed, Nadia G. Abd El-Fattah.

**Resources:** Mohamed S. Abd El-baki, Ahmed Elbeltagi, Nadia G. Abd El-Fattah.

**Software:** Mohamed S. Abd El-baki, Mohamed Maher Ibrahim.

**Validation:** Ahmed Elbeltagi.

**Visualization:** Mohamed S. Abd El-baki.

**Writing – original draft:** Mohamed S. Abd El-baki.

**Writing – review & editing:** Mohamed S. Abd El-baki, Mohamed Maher Ibrahim, Salah Elsayed, Ahmed Elbeltagi, Ali Salem, Nadia G. Abd El-Fattah.

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
