## [Decision Letter · Decision Letter 0]

15 Feb 2026

Dear Dr. Salem,

Thank you for submitting your manuscript to PLOS ONE. After careful consideration, we feel that it has merit but does not fully meet PLOS ONE’s publication criteria as it currently stands. Therefore, we invite you to submit a revised version of the manuscript that addresses the points raised during the review process.

We look forward to receiving your revised manuscript.

Kind regards,

Ömer Faruk Coşkun, Ph.D

Academic Editor

PLOS One

Journal Requirements:

Reviewer's Responses to Questions

**Comments to the Author**

1. Is the manuscript technically sound, and do the data support the conclusions?

Reviewer #1: Yes

Reviewer #2: Yes

2. Has the statistical analysis been performed appropriately and rigorously?

Reviewer #1: Yes

Reviewer #2: Yes

3. Have the authors made all data underlying the findings in their manuscript fully available?

Reviewer #1: No

Reviewer #2: Yes

4. Is the manuscript presented in an intelligible fashion and written in standard English?

Reviewer #1: Yes

Reviewer #2: Yes

Reviewer #1: the relationship between RGB image indices and plant indices in tomato plants, along with regression analysis was investigated in this study. The two-year experimental period was involved and the experimental workload is quite heavy. Selection of image indices and plant indices were reasonable and the regression results were credible. However, several issues require clarification: 1. Sample data specifications: Providing sample size, open-field experimental area, and final tomato yield would enhance understanding of the experimental subjects. 2. Methodological rigor: While the workload was substantial, the analytical approach lacks systematic completeness. For instance, in ANOVA, presenting both significance levels and effect size metrics (e.g., correlation coefficients) would help readers assess conclusion reliability and correlation strength between variables, rather than relying solely on significance levels. 3. Methodological innovation: Established models for image-plant index regression should be compared. Does the author's model demonstrate advantages over classical approaches or state-of-the-art methodologies? Comparative analysis is warranted to validate its innovation.

Reviewer #2: Manuscript is well organized. Material and methods are suitable for research purposes. This manuscript is acceptable for publication.There is one correction:

In line 119 CROWAT should be corrected as CROPWAT.

.

Reviewer #1: No

Reviewer #2: No

---

## [Author Response · Author response to Decision Letter 1]

20 Feb 2026

Predicting water status, growth and yield of tomato under different irrigation regimes using the RGB image indices and artificial neural network model

Comments to the Author

1. Is the manuscript technically sound, and do the data support the conclusions?

Reviewer #1: Yes

Reviewer #2: Yes

2. Has the statistical analysis been performed appropriately and rigorously?

Reviewer #1: Yes

Reviewer #2: Yes

3. Have the authors made all data underlying the findings in their manuscript fully available?

Reviewer #1: No

Reviewer #2: Yes

4. Is the manuscript presented in an intelligible fashion and written in standard English?

Reviewer #1: Yes

Reviewer #2: Yes

5. Review Comments to the Author

Reviewer #1 Comment Response

Q(1) The relationship between RGB image indices and plant indices in tomato plants, along with regression analysis was investigated in this study. The two-year experimental period was involved and the experimental workload is quite heavy. Selection of image indices and plant indices were reasonable and the regression results were credible. However, several issues require clarification:

Sample data specifications: Providing sample size, open-field experimental area, and final tomato yield would enhance understanding of the experimental subjects. Thank you for your valuable suggestion. The sample size is detailed in lines 181-197, the open-field experimental area is specified in line 108, and the final tomato yield is presented in lines 307-310.

Q(2) Methodological rigor: While the workload was substantial, the analytical approach lacks systematic completeness. For instance, in ANOVA, presenting both significance levels and effect size metrics (e.g., correlation coefficients) would help readers assess conclusion reliability and correlation strength between variables, rather than relying solely on significance levels. Thank you for your comment. In response, we have added effect sizes to clarify the impact of irrigation regimes on FB, DB, CWC, SPAD, SMC, and yield. These values have been incorporated into the manuscript (lines 261–307). To determine significant differences between means, as presented in Figure 3 and Table 5, Tukey’s post-hoc test was applied at significance levels of P ≤ 0.05, 0.01, and 0.001. Regression models were fitted to examine the relationships between plant traits and RGB image indices, as well as between actual and predicted values from the ANN models. The corresponding correlation coefficients are displayed in Figures 4-7.

Q(3) Methodological innovation: Established models for image-plant index regression should be compared. Does the author's model demonstrate advantages over classical approaches or state-of-the-art methodologies? Comparative analysis is warranted to validate its innovation. Thank you for your comment. The primary objective of this study was not to compare different modeling approaches, but rather to develop a non-destructive method for assessing plant characteristics using field and laboratory measurements as reference standards. Accordingly, the accuracy of the artificial neural network model was evaluated by comparing its predictions against actual field and laboratory values, as presented in Tables 6-8. This approach aligns with our goal of providing a reliable, time-saving alternative to conventional destructive methods that can also support irrigation management decisions.

Reviewer #2 Comment Response

Q(1) Manuscript is well organized. Material and methods are suitable for research purposes. This manuscript is acceptable for publication. There is one correction:

In line 119 CROWAT should be corrected as CROPWAT. Thank you for your comment. The necessary adjustment has been made.

---

## [Editor Report · Decision Letter 1]

11 Mar 2026

Dear Dr. Salem,

Thank you for submitting your manuscript to PLOS ONE. After careful consideration, we feel that it has merit but does not fully meet PLOS ONE’s publication criteria as it currently stands. Therefore, we invite you to submit a revised version of the manuscript that addresses the points raised during the review process.

We look forward to receiving your revised manuscript.

Kind regards,

Ömer Faruk Coşkun, Ph.D

Academic Editor

PLOS One

Journal Requirements:

**Additional Editor Comments:**

Dear Authors,

Thank you for submitting the revised version of your manuscript. The study is potentially suitable for publication and the revision has addressed several of the reviewers’ earlier concerns. However, after editorial assessment of the revised manuscript, I find that a small number of issues still need to be corrected before the manuscript can be accepted. I am therefore inviting a minor revision.

Please address the following points carefully in the next revision:

The irrigation model name is still incorrect in the manuscript text. Although the response letter states that this was corrected, the manuscript still reads “CROWAT” in the irrigation methods section. This should be corrected to “CROPWAT” consistently throughout.

Year / season information must be harmonized throughout the manuscript. The abstract refers to the 2021/2022 and 2022/2023 growing seasons, the Methods section states that the field experiments were conducted during 2022 and 2023, and the biomass sampling paragraph refers to measurements in 2021 and 2022. Please revise these sections so that the experimental timeline is fully consistent everywhere.

The ANN methodology requires clearer technical description. Please specify the actual correlation threshold used for feature selection, clarify exactly which scikit-learn implementation and solver(s) were used, and revise the description of training/testing versus cross-validation so that the workflow is unambiguous. At present, the method description remains difficult to follow in this respect.

Please check and correct the equation given for R². The reported formula appears incorrect/incomplete as presented in the current manuscript and should be verified carefully against the actual metric used in the analysis.

Please remove the remaining language and typographical errors throughout the manuscript. Examples visible in the revised version include “Re-mote,” “pcercentage,” “conent,” “soil moister,” “phybrid,” “Identify” instead of “Identity,” “These observed is,” and “DB wights.” Please perform a final careful proofreading of the full manuscript, tables, and figure legends.

Please revise the interpretation regarding T75 in the Discussion. The manuscript states that T75 was favorable “with no significant reduction in the tomato crop,” whereas the preceding results section states that Tukey post-hoc testing showed significant pairwise differences between treatments for yield. Please reconcile this statement with the actual statistical results and avoid overstating the conclusion.

Once these minor issues are addressed, the manuscript can be reconsidered promptly.

Sincerely.

---

## [Author Response · Author response to Decision Letter 2]

15 Mar 2026

No. Comments Response

1 The irrigation model name is still incorrect in the manuscript text. Although the response letter states that this was corrected, the manuscript still reads “CROWAT” in the irrigation methods section. This should be corrected to “CROPWAT” consistently throughout. We appreciate your comment and have revised the text accordingly.

2 Year / season information must be harmonized throughout the manuscript. The abstract refers to the 2021/2022 and 2022/2023 growing seasons, the Methods section states that the field experiments were conducted during 2022 and 2023, and the biomass sampling paragraph refers to measurements in 2021 and 2022. Please revise these sections so that the experimental timeline is fully consistent everywhere. Thank you for your feedback. We have made the requested adjustments.

3 The ANN methodology requires clearer technical description. Please specify the actual correlation threshold used for feature selection, clarify exactly which scikit-learn implementation and solver(s) were used, and revise the description of training/testing versus cross-validation so that the workflow is unambiguous. At present, the method description remains difficult to follow in this respect. We thank the reviewer for this comment. The ANN methodology has been revised as suggested (see lines 202-255).

4 Please check and correct the equation given for R². The reported formula appears incorrect/incomplete as presented in the current manuscript and should be verified carefully against the actual metric used in the analysis. Thank you for your feedback. We have corrected the equation for R² as requested.

5 Please remove the remaining language and typographical errors throughout the manuscript. Examples visible in the revised version include “Re-mote,” “pcercentage,” “conent,” “soil moister,” “phybrid,” “Identify” instead of “Identity,” “These observed is,” and “DB wights.” Please perform a final careful proofreading of the full manuscript, tables, and figure legends. We thank the reviewer for this feedback. As requested, we have corrected the remaining language and typographical errors throughout the manuscript.

6 Please revise the interpretation regarding T75 in the Discussion. The manuscript states that T75 was favorable “with no significant reduction in the tomato crop,” whereas the preceding results section states that Tukey post-hoc testing showed significant pairwise differences between treatments for yield. Please reconcile this statement with the actual statistical results and avoid overstating the conclusion. We thank the reviewer for this feedback. We have revised the interpretation regarding T75 in the Discussion, as requested (see lines 343–346).

---

## [Editor Report · Decision Letter 2]

20 Mar 2026

Predicting water status, growth and yield of tomato under different irrigation regimes using the RGB image indices and artificial neural network model

PONE-D-25-50619R2

Dear Dr. Salem,

We’re pleased to inform you that your manuscript has been judged scientifically suitable for publication and will be formally accepted for publication once it meets all outstanding technical requirements.

Kind regards,

Ömer Faruk Coşkun, Ph.D

Academic Editor

PLOS One

Additional Editor Comments (optional):

Dear Authors,

Thank you for your revision. The manuscript has been improved and is suitable for publication.

Please carefully check and correct remaining minor language, grammar, and typographical issues during the proof stage.

Best regards.

---

## [Editor Report · Acceptance letter]

PONE-D-25-50619R2

PLOS One

Dear Dr. Salem,

I'm pleased to inform you that your manuscript has been deemed suitable for publication in PLOS One. Congratulations! Your manuscript is now being handed over to our production team.

Kind regards,

on behalf of

Professor Ömer Faruk Coşkun

Academic Editor

PLOS One